# Inheritance of Secondary Metabolites and Gene Expression Related to Tomato Fruit Quality

**DOI:** 10.3390/ijms23116163

**Published:** 2022-05-31

**Authors:** Estelle Bineau, José Luis Rambla, Renaud Duboscq, Marie-Noëlle Corre, Frédérique Bitton, Raphaël Lugan, Antonio Granell, Clémence Plissonneau, Mathilde Causse

**Affiliations:** 1INRAE, UR1052, Génétique et Amélioration des Fruits et Légumes, 67 Allée des Chênes, Centre de Recherche PACA, Domaine Saint Maurice, CS60094, Montfavet, 84143 Avignon, France; estelle.bineau@gautiersemences.com (E.B.); renaud.duboscq@inrae.fr (R.D.); marie-noelle.corre@inrae.fr (M.-N.C.); frederique.bitton@inrae.fr (F.B.); 2GAUTIER Semences, Route d’Avignon, 13630 Eyragues, France; clemence.plissonneau@gautiersemences.com; 3Instituto de Biología Molecular y Celular de Plantas, CSIC, Universidad Politécnica de Valencia, 46022 Valencia, Spain; jrambla@ibmcp.upv.es (J.L.R.); agranell@ibmcp.upv.es (A.G.); 4Departamento de Ciencias Agrarias y del Medio Natural, Universitat Jaume I, 12071 Castellón de la Plana, Spain; 5UMR Qualisud, UAPV/CIRAD, 84140 Avignon, France; raphael.lugan@univ-avignon.fr

**Keywords:** tomato, breeding, flavour, volatiles, mode of inheritance, gene expression

## Abstract

Flavour and nutritional quality are important goals for tomato breeders. This study aimed to shed light upon transgressive behaviors for fruit metabolic content. We studied the metabolic contents of 44 volatile organic compounds (VOCs), 18 polyphenolics, together with transcriptome profiles in a factorial design comprising six parental lines and their 14 F1 hybrids (HF1) among which were five pairs of reciprocal HF1. After cluster analyses of the metabolome dataset and co-expression network construction of the transcriptome dataset, we characterized the mode of inheritance of each component. Both overall and per-cross mode of inheritance analyses revealed as many additive and non-additive modes of inheritance with few reciprocal effects. Up to 66% of metabolites displayed transgressions in a HF1 relative to parental values. Analysis of the modes of inheritance of metabolites revealed that: (i) transgressions were mostly of a single type whichever the cross and poorly correlated to the genetic distance between parental lines; (ii) modes of inheritance were scarcely consistent between the 14 crosses but metabolites belonging to the same cluster displayed similar modes of inheritance for a given cross. Integrating metabolome, transcriptome and modes of inheritance analyses suggested a few candidate genes that may drive important changes in fruit VOC contents.

## 1. Introduction

The cultivated tomato (*Solanum lycopersicum* L.) is both a model organism to study the physiological and genetic control of fleshy fruit development and composition [1] and the most produced and consumed vegetable worldwide. More than 186 million tons were produced in 2020 [2], with about 75% for the fresh market [3]. The cultivated tomato encompasses both cherry type tomato (*S. lycopersicum* var. *cerasiforme*, ‘SLC’) and big-fruited tomato (*S. lycopersicum* L., ‘SL’). The goals of breeding programs have evolved starting from the 1950s [4]. Increasing interest towards the genetic control of flavour followed consumer first complaints about tomato loss of flavour in the 90’s [5,6]. Concomitantly, the hybrid state has been increasingly exploited in commercialized tomato varieties. This preference is supported by heterosis, also called hybrid vigor, where hybrids show superior agronomical performances compared to both inbred parental lines. In tomato, breeders mostly exploit the hybrid state to combine resistance genes and to prevent the use of their parental lines by competitors. Heterosis is not exploited knowingly for traits other than production stability such as fruit size or shape homogeneity [7], as heterosis remains limited for yield. In 2018, about 89% of tomato varieties were commercialized as F1 hybrids (‘HF1’) in Europe [8]. Tomato organoleptic quality is the sum of fruit physical (aspect and texture) and chemical properties. The metabolite content of tomato fruit contributes to both nutritional value and flavour [9]. While nutritional quality is straightforward to assess with the quantification of vitamins and phytonutrients (phenolic compounds and carotenoids), flavour is much more complex. It relies on the integrated perception of (i) taste, mostly driven by sugars (glucose and fructose) and organic acids (malate and citrate) and (ii) aroma, with no less than 400 volatile organic compounds (‘VOCs’) identified thus far in tomato [10]. The number of VOCs supposed to contribute to overall flavour has been cut down to approximately 30 [11]. Most of the VOCs with active aroma in tomato fruits can be classified into distinct groups depending on the compounds they derive from, namely: (i) fatty acid or lipid derived (L-der); (ii) branched chain amino acids (BCAA-der) closely linked to (iii) sulfur containing compounds (S-der); (iv) apocarotenoids (C-der), a small group of irregular terpenoids derived from carotenoids, sharing part of their precursors with (v) terpenoids (T-der) that comprise both mono- and sesquiterpenes; (vi) benzenoids (B-der) and (vii) phenylalanine derivatives (Phe-der), both deriving from the phenylalanine amino acid precursor but broken down into two distinct metabolic pathways by specialized enzymes [12]. Phenolic compounds, otherwise called polyphenols or non-volatile phenylpropanoids, mostly derive from the shikimate pathway as do B-der and Phe-der VOCs. Increasing reports attribute their antioxidant properties to the prevention of cardiovascular and chronicle diseases in human [13,14].

The genetic architecture of fruit flavour and nutritional quality has been thoroughly studied in homozygous plant material such as Introgression Lines (‘ILs’) and Recombinant Inbred Lines (‘RILs’) derived from biparental populations, or Genome Wide Association Study (‘GWAS’) panels covering the genetic diversity of a species. Hundreds of QTLs [15,16,17,18,19,20] and association signals [11,21,22,23,24,25] have been suggested to control flavour and nutritional quality-related traits. However, while most tomato varieties are commercialized as HF1, the impact of the heterozygous state on flavour-related traits has long been overlooked. [25] performed the first GWAS on a testcross panel to identify associations specific to HF1, or shared between HF1 and line panels for flavour-related traits. The authors suggested seven chromosome regions with clusters of associations simultaneously involved in several key VOCs for tomato aroma, where improvement for flavour-related traits may be more efficient than QTLs only identified in line panels.

Besides the genetic architecture of such traits, knowing the inheritance of metabolic content is highly informative for breeders who seek to improve flavour and nutritional quality in their hybrids and who may exploit QTLs with heterotic effect for these traits. Likewise, phenotypic differences may arise between reciprocal hybrids given the parental line chosen as the female/male parent. In tomato, heterosis for metabolite content has mostly been investigated for primary metabolites such as amino acids, sugars and organic acids [26,27,28,29]. An important additive pattern has been reported for these traits, although several traits exhibited over-dominant or over-recessive patterns. Among the 26 metabolites studied in [28], the authors reported that for a given metabolite, the mode of inheritance changed between crosses and was also variable for the same cross-metabolite but at different fruit development stages. Negative heterosis has, however, been consistently reported for amino acids in maize [30,31], in addition to a consistent positive heterosis pattern for sugars and lipids [31]. More recently, [29] studied a tomato full diallel mating design gathering five parental lines and their 20 hybrids along with reciprocal hybrids evaluated on 12 agronomic traits and 28 metabolites (sugars, organic acids and amino acids). They reported that most metabolites showed positive heterosis in the F1 hybrids, with up to 50% of mean increased content in the hybrids. On the other hand, agronomic traits were more subjected to additivity. Moreover, the authors reported reciprocal effects for 46% of the metabolites they assessed, amino acids being over-represented among metabolites displaying significant difference between reciprocal hybrids. As the mode of inheritance of secondary metabolite content hasn’t been investigated in tomato so far, insight from other species highlights the possibility to exploit heterosis for such traits as well, as shown for capsaicin in *Capsicum anuum* [32], or 34 secondary metabolites assessed in a full diallel mating design gathering seven *A. thaliana* parental lines [33]. The authors reported a more pronounced non-additive mode of inheritance for secondary compared to primary metabolites [33]. On the other hand, a predominant additive mode of inheritance has been reported for gene expression and protein accumulation in maize [30]. Studies on heterosis and cross direction effects on metabolite content demonstrate the utmost importance of considering the right combination and cross direction of parental lines to observe hybrids transgressing parental values for traits of interest. Yet, the mode of inheritance of VOCs and phenolics has been overlooked so far. 

To take full advantage of the hybrid state for flavour and nutritional quality, the present study aims at providing knowledge concerning: (i) the mode of inheritance of tomato metabolite content; (ii) the mode of inheritance of genes underlying metabolite content; (iii) the possible impact of the cross direction on metabolite content. To reach this aim, we studied the modes of inheritance of metabolome and transcriptome profiles in fruit produced following a factorial design comprising six parental lines and their 14 HF1 obtained from four intra- and ten interspecific crosses. We produced five pairs of reciprocal HF1 to investigate the impact of the genetic cross direction on VOC and phenolic content. We integrated metabolome and transcriptome datasets through a correlation network analysis to further characterize the mode of inheritance of differentially expressed genes correlated to key VOCs and phenolic compounds. 

## 2. Results

### 2.1. Trait Variation, Correlations and Heritabilities

We assessed fruit weight, SSC and the abundancy of 44 VOCs, 18 polyphenols, two amino acids and two alkaloids in six lines (two wild ancestor *Solanum pimpinellifolium* –‘SP’ lines, three SLC and one SL), their four intraspecific HF1 (two SP × SP, two SLC × SLC) and ten interspecific HF1 (2 SP × SLC, 4 SP × SL, 4 SLC × SL) (Figure 1). The full names and abbreviations of the traits and secondary metabolites we assessed are presented in Appendix A. Appendix A also indicates which traits and metabolites impact tomato aroma based on the literature. Eight out of the 68 traits studied showed no significant genotype effect in the analyses of variance (Appendix A). These eight traits also had the lowest broad-sense heritability values and showed short range of variation between the 14 crosses (Appendix A). For the other traits, heritability values ranged from 0.24 for the carotenoid-derived NERAL to 0.99 for the alkaloid LYPR, and 50 traits had heritability values higher than 0.4. The cluster analysis on secondary metabolites and SSC of these 20 genotypes was consistent with [25,34] studies. 

To investigate the link between secondary metabolites, SSC and fruit weight, we calculated correlation coefficients between each pair of traits as illustrated in Appendix A. We displayed the content of each metabolite (clustered beforehand) in each accession and for their three replicates (Appendix A) and further validated the cluster analysis (Appendix A). Fruit weight was positively correlated with phenylalanine, benzenoid and terpenoid-derived volatiles and showed significant negative correlations with compounds from all other metabolic pathways (Appendix A). We studied three major classes of secondary metabolites derived from the aromatic amino acid phenylalanine (PHE), namely polyphenolics, benzenoid VOCs and phenylalanine-derived VOCs. These three classes showed contrasted correlations between one another. Phenylalanine-derived VOCs associated with consumer liking had varying correlations with the three classes of polyphenolics studied here and described in [14]. They clustered together with phenylpropanoids such as TRICAFQ and CRYPTOCH (Appendix A) and correlation analysis showed significant positive correlation with these compounds (Appendix A). On the contrary, we found negative correlations between phenylalanine-derived VOCs and flavones such as NARINCH and flavonols such as KAEMGR and RUTIN. On the other hand, we found no significant correlations between benzenoid VOCs negatively correlated to consumer liking and phenylalanine-derived VOCs, but strong negative correlations between benzenoid VOCs and one of their direct precursor CAF and flavonols. As for the group of polyphenolics, we identified two main clusters gathering on the one hand phenylpropanoids, and on the other hand flavones and flavonols (Appendix A). Phenylpropanoids (resp. flavones and flavonols) exhibited positive correlations between one another, but no significant correlation to flavones and flavonols (resp. phenylpropanoids) (Appendix A). 

As for carotenoid-derived VOCs positively associated with consumer liking, we identified distinct clusters. The cyclic apocarotenoids BCYCL and BIONO clustered together in the cluster analysis (Appendix A) with strong positive correlation (0.82). Linear apocarotenoids 6MHON, GRACE NERAL and GERIAL clustered together in a second cluster (positive correlations ranging from 0.74 to 0.93), shared with long-chained fatty acid derived VOCs. BDAM was isolated from the other two apocarotenoid clusters but showed positive correlations, although weaker, with linear apocarotenoids such as GRACE and 6MHON (from 0.32 to 0.59).

We identified two major clusters of fatty-acid-derived VOCs that matched those previously described [25,34]: in the first cluster the C_5_ PENAL and C_7_ to C_10_ lipid-derived VOCs, while in the second the four C_6_ lipid-derived volatiles associated with ‘green’/‘grassy’ aroma in addition with 2EFUR, and further though still in the same cluster two C_5_ and two C_7_ fatty acid derived VOCs (Appendix A).

### 2.2. Modes of Inheritance of Secondary Metabolites and Differences between Reciprocal Hybrids

We analysed the mode of inheritance of secondary metabolites in light of the cluster they belonged to. The mode of inheritance of VOCs, polyphenolics, SSC and fruit weight was assessed within crosses with significant differences for these traits. In 56% of the cases (crosses and traits), no significant differences could be found within crosses. Additivity (‘A’) was detected in 19% of the cases, then recessiveness (‘R’) and over-recessiveness (‘OR’) (10% and 8%, respectively) and finally over-dominance (‘OD’) and dominance (‘D’) patterns in 5% and 2% cases, respectively.

We observed transgressions beyond and below parental values for 47 traits relevant to breeding among which SSC, 23 VOCs impacting tomato overall aroma, and polyphenols with antioxidant properties. Except for the terpene-derived LIN and the polyphenol 35DICAFQ that we found both OD and OR among the 14 crosses, transgressions were always of a single type among the 14 crosses for the 47 traits. 

Crosses involving SP1 and a parent from a different species displayed the highest number of significant differences across traits (Figure 2a). When the mode of inheritance could be assessed, SP1 crosses were mostly found OR or R. Consequently, they were the crosses with the highest number of these patterns. Moreover, we often identified consistency in the mode of inheritance of a given trait between crosses sharing the SP1 parent.

Only few traits exhibited consistency of their mode of inheritance across the 14 crosses. Fruit weight was additive for 11 out of 14 crosses, while crosses between SP1 and SP2 led to over-dominance. Phenylalanine-derived PHENE was additive for eight out of 11 crosses displaying significant differences, and the polyphenolic RUTIN5O was found additive in the 11 crosses showing a significant genotype effect for the trait. Crosses involving SLC2 and SL (SL being the only big-fruited accession) were those with the highest content in benzenoid and terpenoid-derived volatiles. Looking at the inheritance of benzenoid VOCs, we found mostly OR to R patterns, with SL highest content in these VOCs always overridden by the other parent value in the resulting HF1. The benzenoids SALI, GUAIA, MESA and EUGEN were mainly found OR to R whichever the cross. Terpenoid-derived VOCs were OR in every cross that involved SP1 and additive in the other crosses. In addition, SSC was D to OD when we crossed SL with an SLC.

Apart from the benzenoid and terpenoid-derived VOCs, no clear pattern could be identified at the metabolic pathway scale. However, when we analysed groups of volatiles in light of the clustering analysis (Appendix A), specific modes of inheritance appeared explicitly: first, traits that belong to the same cluster have consistent modes of inheritance for a given cross; second, except for reciprocal hybrids and crosses involving the SP1 parent, we hardly identified consistent modes of inheritance from one cross to another, except for reciprocal crosses. 

We split the large family of fatty acid volatiles into two distinct clusters that displayed distinct modes of inheritance. On the one hand the cluster consisting mostly of C_6_ lipid-derived VOCs showed mainly OR to R patterns for SP crosses and no differences for other crosses. On the other hand, long chain lipid-derived VOCs had OD patterns in crosses that involved one or two SLC. As in cluster analysis, we found three distinct groups of carotenoid derived VOCs with specific inheritance patterns: BDAM mode of inheritance closely matched that of SSC with D to OD patterns in crosses involving parental lines from different species. When we assessed significant differences, BCYCL and BIONO showed consistent additivity patterns for a given cross and the remaining linear apocarotenoids were also mostly additive. 

We compared trait values between each of the five pairs of reciprocal HF1. We found significant differences between reciprocal hybrids for seven metabolites, among which two VOCs correlated to consumer liking (MESA and BDAM) and three polyphenolics (NEOCHL, DIFER and 34DICAFQ). Only MESA mode of inheritance changed from “OR” to “A” between reciprocal crosses with a lower content in MESA when SLC2 was used as a female rather than a male parent in SLC2 × SL crosses. Otherwise, modes of inheritance were consistent despite significant differences between reciprocal HF1 values. Although both crosses gathering SL and SLC2 resulted in OD mode of inheritance for the carotenoid-derived BDAM positively correlated to consumer liking, we found higher content of the VOC when SL was used as the female parent. Both polyphenolics NEOCHL and 34DICAFQ showed higher content when we used SP1 as the female parent in SP1 × SL crosses. The polyphenolic DIFER displayed higher content in the HF1 with SLC2 as the female parent in SLC2 × SL crosses. 

### 2.3. Differential Gene Expression between Parental Genotypes and HF1 and Inheritance Patterns of Gene Expression

To identify genes whose expression is impacted by a genotype effect, we first filtered genes with low expression between the parental lines and 14 HF1. The quality control filters kept 13,127 genes. The first two components of the PCA performed on this subset accounted for 37% of the total variation of these genes (Appendix A). The genotype factor appeared central to this variation, while no pattern of replicate effect appeared in the PCA. We then performed a differentially expressed genes (‘DEG’) analysis on the subset of 13,127 genes and worked on all possible comparisons between the 20 genotypes studied in the factorial design. As a result, 12,896 genes were differentially expressed (‘DE’) in at least one comparison. Thus, more than 95% of all the genes mapped were DE in at least one condition. The remaining 231 genes not DE showed no significant enrichment in gene ontology (‘GO’) terms. 

We first looked at the number of DEGs found between the parental lines of each cross and compared that number to the genetic similarity calculated between parental lines based on 7442 Illumina SNP markers. While we expected the most genetically distant parental lines to display the highest number of DEGs, we found three pairs of parental lines that did not verify that hypothesis (Appendix A). Although SL and SP2 were the most distant parental (Figure 2b) lines in the factorial design (42% marker similarity), they came third in number of DEGs. Moreover, contrary to assumptions based on the domestication history of tomato, we found that SLC1 and SLC3 were more genetically distant than the wild tomato relative SP1 and the big-fruited SL. SLC3 and SLC1 were also the two parental lines with the highest number of DEGs -6966 DEGs-. On the other hand, we found the lowest number of DEGs between SP1 and SP2 -374 DEGs-, although they shared only 64% of marker similarity. We observed that the more DEGs we found between parental lines, the higher the number of metabolites with significant genotype effect within the trio of parental lines and HF1 suggesting that these DEGs drive the trait variation within the trio. 

We assessed the mode of inheritance of the 12,896 DEGs. The frequency of each pattern was close to what we found for metabolic traits. First came no significant expression difference within crosses (‘ns’, 61%). We then found additivity in 20% of the cases, over-dominance and over-recessiveness both in 6% of the cases and finally recessiveness and dominance in 4% and 3% of the cases, respectively. Comparing Figure 2a,b, we observed roughly similar frequencies of modes of inheritance between transcriptome and metabolome for a given cross. Again, the cross between SP2 and SP1 and its reciprocal cross showed the lowest number of significant differences in gene expression among the 12,896 DEGs. When found significantly different, these DEGs were mostly over-dominant or over-recessive while in the other crosses, additivity was the main pattern (Figure 2b) as was also seen for the metabolome dataset (Figure 2a). 

We found differences in gene expression between three out of the five pairs of reciprocal HF1, with eight to 15 DEGs found between reciprocal HF1 (Appendix A). Among the 30 unique gene IDs, three DEGs (Solyc02g069110- ‘Cathepsin B-like cysteine proteinase’; Solyc02g071810 – ‘Protein kinase domain’; Solyc03g097700 – ‘O-methyltransferase’) were commonly found in the contrasts [SL × SP2-SP2 × SL] and [SP1 × SP2-SP2 × SP1] and one DEG (Solyc03g116930 – ‘Phospholipase-like protein’) was shared between the contrasts [SL × SP2-SP2 × SL] and [SL × SP1-SP1 × SL].

### 2.4. Integrating Metabolome and Transcriptome to Identify the Genes Underlying Key Metabolites Synthesis in Each Metabolic Pathway

To gain insights into the genes related to the synthesis of the metabolites, we constructed gene co-expression modules using the matrix of 12,896 DEGs with WGCNA. We identified 12 co-expression modules, with 39 (green module) to 686 (turquoise module) genes comprised within each module (Figure 3a). We then assessed the correlations between each trait and each module eigengene (first principal component of a given module). Only significant correlations are displayed in Figure 3a. The correlation patterns followed the metabolite clusters highlighted in Appendix A. Correlations ranged from −0.83 (P_RUTIN5O - red module) to 0.94 (P_PANTO – salmon module). Metabolites belonging to the same clusters displayed similar correlations to each module. Likewise, we found back the antagonisms highlighted in Appendix A between metabolites. For instance, we reported negative correlations between benzenoid VOCs and the two classes of polyphenolics flavones and flavonols. We found these same antagonisms with the transcriptome dataset with opposite correlations to the same modules between flavones (NARIN7OG) and flavonols (KAEMGR - KAEM3RUT) on the one hand, and benzenoid VOCs (EUGEN – GUAIA – MESA – SALI) on the other hand.

To verify whether modules showed significant enrichment in the different pathways we studied, we computed GO term enrichments. We found eight co-expression modules with significant enrichment in GO terms (Appendix A). Among them, the blue module (n = 482 genes) was enriched in several GO terms related to glycosyl transferase activities (*p*-value = 3.20 × 10^−7^ for the most significant) which play key roles in the synthesis of benzenoid VOCs. The module showed the highest positive correlations with benzenoid VOCs MESA, GUAIA EUGEN and SALI (from 0.42 to 0.66). The turquoise module (n = 686 genes) was enriched in GO terms related to cellular components, namely ‘photosystem’ and ‘photosystem I’ (*p*-value = 4.2 × 10^−12^ and 4.7 × 10^−8^, respectively), ‘photosynthetic membrane’ (4.2 × 10^−12^), the most significant being ‘thylakoid’ (*p*-value = 2.9 × 10^−13^). The turquoise module showed positive correlations with linear apocarotenoids such as 6MHON, GRACE or NERAL (0.67, 0.65 and 0.47, respectively). Synthesis of these VOC precursors, among which lycopene, takes place in the chloroplast and further enzymatic activity depends on light and photosystems [35]. This co-expression module also exhibited strong positive correlations with flavones (NARIN7OG, 0.68) and flavonols (KAEM3RUT, 0.69).

As part of the integration analysis, correlations were computed between the genes within each module and the different metabolites. We present in Appendix A the metabolite-gene significant correlations when the biological function of the gene has previously been proposed to play a role in the metabolite synthesis. In the turquoise module, we found the two carotenoid cleavage dioxygenase cloned genes *CCD1A* and *CCD1B* correlated with GRACE and 6MHON (correlation = 0.62 and 0.74, respectively) and 6MHON (0.56), respectively. For phenolic compounds, we found the cloned gene *4CL,* which is annotated as a ‘4-coumarate:CoA ligase’. This gene catalyses the last step of the overall phenylpropanoid pathway before the pathway is ramified into flavones or non-volatile phenylpropanoids [36]. We found strong positive correlations between the gene expression and seven phenolic compounds (up to 0.9 for CHLOROG, *p*-value = 5.1 × 10^−23^). Within this module, we also identified Solyc01g102950 annotated as a ‘Lycopene beta/epsilon cyclase protein’ and not yet cloned as a QTL, with a correlation of 0.58 (*p*-value = 1.01 × 10^−6^) with 6MHON. In the blue co-expression module, we identified three glycosyltransferase genes (Solyc08g006330, Solyc10g079950 and Solyc11g013490) whose expressions were correlated to the contents of four benzenoid VOCs (MESA, GUAIA, EUGEN and SALI) for Solyc08g006330 and Solyc10g079950, while Solyc11g013490 expression was correlated to three benzenoid VOCs (EUGEN, GUAIA and MESA). We found a negative correlation between Solyc08g006330 expression and the four benzenoid VOCs (down to −0.66 for SALI, *p*-value = 7.03 × 10^−9^), while the two other genes displayed positive correlations for the VOCs they were correlated to (up to 0.68 for Solyc08g006330-SALI, *p*-value = 7.03 × 10^−9^). When we looked at the mode of inheritance of the four benzenoid VOCs in light of the mode of inheritance of Solyc08g006330, we identified consistency (with opposite effect): MESA exhibited over-recessiveness in three crosses where Solyc08g006330 was over-dominant (Figure 3b). Figure 4 displays linear regressions of the four benzenoid VOCs according to Solyc08g006330 expression. The highest the gene expression, the lowest the corresponding metabolite content with coefficients of determination ranging from 0.58 (GUAIA) to 0.83 (EUGEN).

## 3. Discussion

Despite the considerable knowledge accumulated over the last 20 years about the genetic architecture of VOCs and phenolics, breeders still need insight into the improvement they can expect in HF1 varieties given the right parental combinations for these traits. To date, most of the efforts towards the characterization of the impact of heterosis and cross direction have been directed on yield-related traits [37,38] and quality traits [26,27,28,39] in tomato. The most exhaustive study addressing the evolution of metabolite content in HF1 compared to parental lines has been carried out on 28 primary metabolites among which amino acids, organic acids and sugars [29]. Studying a factorial design comprising intra, inter and reciprocal crosses, we characterized the modes of inheritance of VOCs and phenolics compounds and identified candidate genes through integrated analysis of transcriptome and metabolome datasets. We thus focused the discussion on several candidate genes and their modes of inheritance. We also investigated maternal effects for both metabolites and gene expression.

### 3.1. The Factorial Design as a Prime Plant Material to Study Variation in Metabolites and Gene Expression

Our experimental design comprised six parental lines and 14 HF1 assessed over three biological replicates. We found significant genotype effect (*p*-value < 0.05) for 28 out of 33 VOCs impacting tomato aroma (Appendix A) and for all phenolic compounds. We also assessed high heritabilities when considering the three biological replicates with 17 out of 18 phenolics exhibiting h^2^ > 0.4 and 20/33 VOCs with h^2^ > 0.4. These results are consistent with those reported in [25]. From the metabolic diversity observed within the trial, we found back metabolic pathways previously identified and support still pending biological hypotheses as to the origin of several VOCs impacting tomato aroma such as the carotenoid-derived BDAM or lipid-derived 2EFUR [34]. 

For the carotenoid-derived VOCs, we found similar results as in [25] where three distinct clusters appeared for carotenoid-derived VOCs. BDAM, though suggested to derive from β-carotene [18], didn’t correlate with BCYCL and BIONO in the factorial design. These two compounds clustered together in Appendix A and derive from β-carotene [40]. BDAM was isolated from the other two apocarotenoid clusters but showed positive, though weak, correlations with linear apocarotenoids such as GRACE and 6MHON. This supports the hypothesis in [34] that BDAM precursor is not β-carotene and that its synthesis depends on a different mechanism than that described for linear apocarotenoids, as reviewed in [41]. The lipid-derived 2EFUR on the other hand belonged to a cluster gathering C_5_ and C_6_ lipid-derived VOCs, as previously found [25,34]. It has been suggested [34] that Z3HEX may be the precursor of 2EFUR, as the metabolic pathway leading to this VOC hasn’t been elucidated yet. Considering the wide diversity of phenylalanine derived compounds that we quantified within this trial, we observed clusters of compounds whose metabolic pathways are now well known [14,42]. We here provide insight into the correlations at stake between the different classes of phenylalanine-derived metabolites. Both phenylalanine and benzenoid VOCs exhibited negative correlations with flavonols such as KAEMGR and RUTIN, but phenylalanine-derived VOCs were positively correlated to non-volatile phenylpropanoids (up to 0.52 between CRYPTOCH and PHENE, *p*-value = 1.85 × 10^−5^). These correlation patterns are consistent with the two clusters of phenolics identified with on the one hand, non-volatile phenylpropanoids and on the other hand, flavones and flavonols (Appendix A). 

To bridge the gap between metabolite content and gene expression, we carried out RNA-sequencing followed with DEG analysis. From the 13,127 genes that passed the quality control filters, more than 98% showed significant differential expression between at least two individuals, the genotype factor being central to the expression variations observed (Appendix A). We computed a co-expression network analysis between the 12,896 DEGs and identified 12 different co-expression modules. Two modules exhibited significant enrichment in biological functions associated with key VOCs and phenolics (Appendix A). We computed correlations between gene expression and metabolite content and suggested candidates when their biological function has previously been proposed to play a role in the metabolite synthesis. Apart from the cloned genes *CCD1A*, *CCD1B* and *4CL* that participate to the synthesis of carotenoid-derived VOCs and phenolics, respectively, we also identified candidates already suggested by [27]. Among the 91 candidate genes correlated to at least one metabolite, nine were also reported by these authors following GWAS analyses of line and test cross panels. Moreover, three of them presented a significant cis-regulation pattern according to an ‘expression GWAS’ performed by [23]. Integrating multi-omics datasets aims at combining evidence pointing in the same direction to better pinpoint causal genes. In that respect, we suggest Solyc08g006330 annotated as a ‘UDP-xylose phenolic glycosyltransferase’ as a candidate gene related to benzenoid VOCs: (i) This gene belongs to the blue module significantly associated with ‘glycosyltransferase’ terms; (ii) we found significant negative correlations with the four benzenoid VOCs (GUAIA, MESA, SALI and EUGEN) whose synthesis depends on glycosyltransferase genes; (iii) this gene was comprised within the Quantitative Genomic Regions defined by [43] for phenylalanine and benzenoid VOCs and [25] reported significant association of the gene with BENZA, another benzenoid VOC; (iv) a significant cis-regulation pattern was reported by [23]. Based on the negative correlations we reported, further work will be necessary to investigate the possible conjugation mechanism of benzenoid VOCs underlined by this gene as conjugation of such compounds prevent their release upon ripening [44]. 

### 3.2. Relevance of Metabolite Clusters to Get Better Insight into Their Modes of Inheritance

When considering the whole metabolome dataset for all crosses, additivity was the major mode of inheritance for metabolite content (19%) once we accounted for not significant differences (56%) within crosses. However, the frequency of non-additive modes of inheritance exceeds that of additive patterns. Predominant non-additive modes of inheritance for metabolite content have previously been reported in tomato [26,27,29] or *A. thaliana* [33] for instance. In the trial, 47 metabolites (69%) with significant effect on aroma or impacting nutritional quality exhibited transgressions, being either over-recessive or over-dominant in the HF1 compared to parental line values. Over-dominance and over-recessiveness accounted to 15% of the modes of inheritance assessed in the factorial design. Moreover, when transgressions were observed for a metabolite, 96% were of a single type for the metabolites here discussed: for a given metabolite, transgressions were either over-recessive or over-dominant whatever the cross. We hardly found consistency in the mode of inheritance of a trait across the 14 crosses. Benzenoid SALI, GUAIA, MESA and EUGEN were mainly found over-recessive or recessive whatever the cross. Fruit weight was additive in 11 crosses, which is consistent with previous findings reporting no heterosis for fruit weight in tomato [28,37]. However, we found that both crosses involving the two SP accessions led to over-dominance. The phenylalanine-derived PHENE and flavonol RUTIN5O were mostly found additive. Except for these traits, no consistent pattern appeared among the crosses. We thus looked at the mode of inheritance in light of the species involved in the 14 crosses. We found that terpenoid-derived VOCs were over-recessive in every cross that involved SP1 and additive in the other crosses. SSC was dominant to over-dominant when we crossed SL with an SLC. Finally, we looked at modes of inheritance based on the metabolic clusters highlighted in Appendix A. Strongly correlated volatiles or phenolics exhibited similar modes of inheritance for a given cross. Rather than whole pathways, considering the inner branching of metabolic pathways with the example of lipid-derived VOCs or compounds derived from phenylalanine helps drawing hypotheses as to the mode of inheritance of metabolites. As such, it would be possible to reduce the number of metabolites quantified to focus on increasing the number of crosses evaluated to identify potential transgressions of interest. Except for amino acids, few studies have reported heredity pattern trends for classes of compounds. In maize, negative heterosis has been reported for most amino acid contents [30,31], which is consistent with our analysis of the two amino acids phenylalanine and tryptophane found both recessive and over-recessive. 

We assessed the mode of inheritance of the 12,896 DEGs identified within the factorial design. The overall frequencies of the modes of inheritance were close between the metabolome and DEG datasets. Gene expression was mostly additive (20%) as reported in maize [30], while other modes of inheritance accounted to 19%. Moreover, the frequencies of the modes of inheritance between metabolome and DEGs were close for a given cross. Furthermore, the more DEGs were found between parental lines, the higher the number of metabolites exhibiting a significant difference at the cross level. As we assumed that some DEGs drive the metabolome variations, we investigated the mode of inheritance of candidate genes in light of the mode of inheritance of the metabolite they were correlated to. If a strong correlation is found between gene expression and metabolite content, breeders might perform quantitative PCR over the gene of interest or, if a polymorphism is identified nearby, follow the marker through MAS schemes. We suggested Solyc08g006330 for its potential role in preventing benzenoid VOC accumulation. When looking at its mode of inheritance, we found that crosses in which the gene expression was over-dominant were those where the mode of inheritance of MESA and SALI were over-recessive (Figure 3b). Moreover, we found that all four benzenoid VOCs showed decreasing content according to increasing expression of the gene (Figure 4) which further points to the key role this gene may play in the subsequent synthesis of benzenoid VOCs. 

### 3.3. Consequences and Application for Breeding F1 Tomato Varieties with Improved Flavour and Nutritional Quality

Knowledge about non-additive modes of inheritance for metabolic content is of the utmost importance to breeders: as the agronomic value of a variety will always rank first among breeding program priorities, getting insight into the possible improvement for flavour and nutritional quality will save time and expenses. 

Although several crops such as rice exhibit strong correlation between heterosis occurrence and genetic distance between parental lines [45,46], we report poor correlation between them within the factorial design, as did [29] with the full diallel design they studied for 28 metabolite contents, or [47]. Indeed, the highest number of transgressions we observed for metabolite content (being over-recessive or over-dominant) was reported for the cross SLC3 × SP1 (12%) gathering parental lines with 41% marker difference while we found 7% of transgressions within the SL × SP2 cross gathering the most dissimilar parental lines (58% dissimilarity). Likewise, SLC3 × SP1 exhibited the highest number of transgressions for DEG expression in the HF1 (19%). Moreover, the number of DEGs between parental lines was poorly correlated to their genetic distance. Therefore, we do not expect more heterosis for metabolite content when crossing genetically distant parental lines. However, we report several consistent patterns for transgressions in the factorial design: first, among the 47 metabolites participating in tomato flavour or nutritional quality and exhibiting transgression in HF1, 45 showed either over-dominance or over-recessive patterns among the 14 crosses; second, the four benzenoid VOCs negatively correlated to consumer liking were found over-recessive or recessive in more than 85% of the cases when significant differences were found. Otherwise, we found additivity. This implies that the big-fruited parental line SL highest content in these compounds was mostly overridden by the small-fruited parental lines with lower content of benzenoid VOCs. Thus, crossing a cherry type and a big-fruited line tomato may result in an aromatic profile close to the cherry type tomato with lower content in such compounds in the HF1. On the other hand, we didn’t study crosses involving two big-fruited lines where different patterns may be observed. To date, numerous transgressions have been reported for tomato non-volatile metabolite content [26,27,28,29]. We extend this knowledge to flavour and nutritional quality-related compounds, which are the target of improvement programs.

With the five pairs of reciprocal hybrids, we investigated the impact of the cross direction on metabolite content and gene expression. We report only seven significant differences between reciprocal HF1 metabolic content, three of which for VOCs correlated to consumer liking. We observed an over-recessive mode of inheritance when the cherry type parent was used as a female in the SLC2 × SL cross and additivity when SL was the female parent. Choosing SL as the female parent in SL × SLC2 crosses resulted in higher content in the carotenoid-derived VOC BDAM positively correlated to consumer liking. Three phenolics also exhibited significant differences and choosing a small-fruited parent as female instead of the big-fruited SL was more beneficial for increasing the corresponding metabolite contents. We looked for DEGs between reciprocal hybrids that may drive the significant differences observed at the metabolic level. First, the lowest number of DEGs between the contrasts we considered was found for reciprocal HF1 with zero to 15 DEGs identified. None of these DEGs were correlated to the metabolites displaying significant differences between reciprocal hybrids. However, three DEGs were consistently found among several pairs of reciprocal HF1. Though we didn’t correlate them to metabolite content variation between reciprocal HF1, they may influence other important traits and thus require further investigation. While 46% of the metabolites [29] investigated displayed significant variations according to the cross direction, we report such differences for only 10% (7/68) of the metabolites, with only MESA inheritance pattern changing due to this difference. Thus, we would suggest investing more efforts towards the achievement of transgressions based on the patterns and the metabolic clusters we highlighted. Cross direction did not appear to affect VOC and phenolic content as much as it did on amino acids, sugars and organic acids as reported in [29]. 

Moreover, since we found consistency in the mode of inheritance of metabolites belonging to the same clusters, transgressions may be achieved for several compounds concomitantly. For phenylalanine-, carotenoid- and benzenoid VOCs displaying consistent correlations to consumer liking within their own metabolic pathway (positive, positive and negative, respectively), this finding may help breeders reduce the number of VOCs to target while increasing the number of crosses investigated as, for a given cross, consistent modes of inheritance and thus increase/decrease of the corresponding metabolites may be achieved. Finally, as we found positive correlations between phenylalanine-derived VOCs and phenylpropanoids, improving both flavour and nutritional quality may be achieved given the identification of common QTLs between the two classes of metabolites, with careful attention to not increase benzenoid VOCs. However, using small-fruited parental lines may prevent increase of benzenoid compounds, although further investigation is necessary. 

## 4. Materials and Methods

### 4.1. Plant Material

We studied a factorial design comprising six parental lines and 14 F1 hybrids (HF1). Among the parental lines, two were tomato wild relatives *S. pimpinellifolium* –“L. pimpinellifolium atypique, site 10” (‘SP1’) and LA1589 (‘SP2’)-, three were cherry type tomatoes –Cervil (‘SLC1’), Stupicke Polni Rane (‘SLC2’) and LA1420 (‘SLC3’)- and the last one was a big-fruited line *S. lycopersicum* L. var. *lycopercisum* –Ferum TMV (‘SL’)- also used as a common tester in GWAS panels in [25]. We produced four intraspecific (two SP × SP and two SLC × SLC) and ten interspecific HF1 (two SP × SLC, four SP × SL and four SLC × SL (Figure 1). Although SL and SLC both belong to the *S. lycopersicum* species, we considered their HF1 as interspecific to ease the reading. Ten HF1 resulted from five reciprocal crosses: SP1 × SP2, SL × SP1, SL × SP2, SL × SLC2, SL × SLC3 were studied both ways. 

### 4.2. Growth Conditions and Fruit Sampling

Plants from the factorial design were grown from April to July 2019 in a plastic greenhouse under soilless and passive irrigation condition on the experimental site of the seed company Gautier Semences in Eyragues, France. Five plants per genotype were cultivated.

Three harvests of red ripe fruits, representing three replicates, were conducted during three consecutive weeks, starting from the 2nd truss for the first harvest and finishing around the 5th truss for the 3rd harvest. At least ten fruits were harvested from each plot and each harvest (up to 30 fruits per harvest for SP accessions). The harvested fruits from each plot were divided in two pools. With the first pool of fruits, we measured an average fruit weight (‘weight’) before crushing the fruit pericarps to measure Soluble Solid Content (‘SSC’ in degree Brix). For the second pool, the pericarp from at least five fruits per plot was flash frozen in liquid nitrogen, ground to powder with a cryogenic mill, and then stored at −80 °C. The original powder was subsampled into three vials destined to (i) VOCs profiling; (ii) quantification of polyphenolic compounds; (iii) RNA-seq analysis. Phenotypic values are available in Appendix A.

### 4.3. VOC Profiling by Gas Chromatography/Mass Spectrometry

Profiling of volatile compounds was performed at the Instituto de Biología Molecular y Celular de Plantas, at the Universidad Politécnica de Valencia, Spain, following the protocol described in [25]. A homogenate comprising the three replicates of the 20 genotypes was injected before and after each batch of the GC-MS as a reference for correction of instrumental drift and fiber aging, in a daily basis. The abundancy of a given VOC in a sample was expressed as the ratio between the VOCs in the sample and the amount detected on that batch in the tomato homogenate.

### 4.4. Phenolic Compound Quantification via UPLC-DAD-ESI-TQ Analysis

Phenolic compounds were quantified on the Avignon metabolomic platform. All solvents were analytical or LC/MS grade. Twenty mg of frozen-dried pericarp were extracted with 1.36 mL of methanol acidified with formic acid (0.1%; v/v). 5 µg of Taxifolin was added as internal standard. The samples were homogenized with a vortex, placed in a water bath at 70 °C for 10 min. During the treatment, the samples were homogenized twice with a vortex. Then, 0.44 mL of ultrapure water were added and the extraction was performed as before. The extract was centrifuged at 5300 rcf for 5 min at 4 °C and the supernatant was filtered through a 0.2 µm membrane and stored at −20 °C until analysis.

UPLC was carried out using an Acquity UPLC Class I (Waters, Milford, MA, USA). Chromatographic separation was achieved on a reversed-phase column, BEH C18 1.7 µm 2.1 × 100 mm, equipped with a guard column (Waters, USA). The mobile phase, consisting of water with formic acid (0.1%, v/v) (eluent A) and acetonitrile with formic acid (0.1%, v/v) (eluent B), was pumped at 0.4 mL/min. The following gradient was applied: 0–0.5 min, 2% B, 0.5–6.5 min, 44.5% B, 6.5–7.5 min, 100% B, 7.5–9 min, 100% B. Then the concentration of B decreased to 2% and the column was equilibrated before the next injection (total run time 10 min). The column temperature was kept at 35°C and the samples at 10 °C. 1 µL was injected. The system was controlled by MassLynx version 1.2 (Waters, Milford, MA, USA). The UPLC system was coupled with a triple quadrupole mass spectrophotometer TQ-XS XEVO (Waters, Milford, MA, USA) working with an electrospray source (ESI) at 150 °C in negative ion mode. Mass acquisitions were performed in scan mode and MRM mode. The MassLynx software controlled the MS analyser. We quantified 20 polyphenolic compounds in addition to two alkaloids as part of the untargeted analysis. Major metabolites were identified according to their MS/MS data with pure standard. Other compounds were identified by comparing their MS/MS data with published data. UPLC-DAD-ESI-TQ data were processed with the TargetLynx software (Waters, USA) to integrate peaks in MRM mode. Results were expressed in area of peak per gram of fresh matter.

### 4.5. RNA Extraction

We performed RNA extraction on the three biological replicates obtained from pools of ripe tomato pericarp. For each sample, total RNA was extracted using the “Spectrum Plant Total RNA” Kit (Sigma-Aldrich, St. Louis, MO, USA) following the manufacturer’s protocol and treated for 15 mn at 20 °C with “On Column DNAse Digestion Set” (Sigma-Aldrich) to remove genomic DNA traces. After extractions, RNA purity was assessed on Nanodrop 1000 (Thermo Fischer Scientific, Waltham, MA, USA); all ratios (A260/280 and A260/230) were comprised between 1.8 and 2.2. RNA integrity was assessed on Bioanalyzer 2100 (Agilent, Santa Clara, CA, USA) using RNA Nano 6000 kit; all RNA integrity numbers (RIN) were between 7.7 and 9.2 and samples were not degraded. RNA concentration was assessed on Qubit 3.0 Fluorometer (Thermo Fischer Scientific, USA) using Qubit RNA Broad Range Assay Kit.

### 4.6. RNA Sequencing, Data Processing and Analysis

Library construction and sequencing (100 bp paired-end strand) of the 60 samples were subcontracted to BGI Genomics. The minimal, maximal, and average amounts of raw sequencing data per sample were estimated to be 2.05 × 10^9^ bp, 2.72 × 10^9^ bp and 2.54 × 10^9^ bp, respectively. Raw sequencing data quality was assessed using FASTQC v.0.11.8 software [48] and aggregated with MULTIQC v.1.7 [49]. Sequences were trimmed using FASTP v.0.20.0 [50]. On average, cleaning steps removed 4.03% of the data (min = 4.02%, max = 4.04%). Remaining data were aligned onto the tomato reference genome (Heinz 1706, v.4.0.0 available on https://solgenomics.net (accessed on 27 April 2022)) and raw read counts per gene were generated for each library using STAR v.2.7.3 with two passes and providing the tomato gene model (annotation v4.0) to support the mapping process. Alignments were filtered to keep only concordantly mapped reads using SAMtools v.1.9 [51]. 

On average, 95.8% of read pairs were uniquely mapped per library (min = 91.7%, max = 96.72%) and 2.1% multi-mapped (min = 1.6%, max = 5.4%). We filtered out genes mapping on chromosome 0 and performed quality control, read count normalization and sampling of genes expressed in the experiment using the workspace *DiCoExpress* with recommended parameters [52]. A total of 13,127 genes remained after the quality control procedure (39% of all detected genes). We performed the DEG analysis on this subset using the workspace *DiCoExpress* with recommended parameters. After normalization, a principal component analysis (PCA) was performed to assess the diversity of the transcriptome data in the 6 lines and 14 F1 hybrids. Then DEGs were detected for all possible contrasts -190- between the 14 HF1 and six parental lines using the *edgeR* package: first, we detected genes with significant expression differences between the 20 genotypes; second, we identified genes with significant expression differences between the five pairs of reciprocal F1 hybrids. The *p*-values were corrected for multiple comparisons using the false discovery rate [53] using a global threshold of 0.05. For further explanations, we refer to the manual of *DiCoExpress*.

### 4.7. Data Processing and Statistics

We used the R software v.3.6.2 [54] to perform statistical analyses and data processing. We first performed a fixed effect analysis of variance with the *car* package on the 66 secondary metabolites, SSC and fruit weight to test for genotype effect with the following model:*y*_*ij*_ = *μ* + *g*_*i*_ + *r*_*j*_ + *ε*_*ij*_,(1)
where *y_ij_* is the trait value of genotype *i* in harvest *j*, *μ* is the intercept, *g_i_* and *r_j_* represent the fixed effects of the genotype and the harvest, respectively, and *ε_ij_* the residual effect. Broad-sense heritability (*h*²) was then computed for every trait with the *lme4* package by using the following linear mixed model:*y**_ij_* = *μ* + *g_i_* + *r_j_* + *ε_ij_*,(2)
where *y_ij_* is the trait value of genotype *i* in harvest *j*, *μ* is the intercept, *g_i_* the random effect of genotype *i*, *r_j_* the fixed effect of harvest *j*, and *ε_ij_* the random residual effect. Then heritability was derived from the variance components of the model as:*h*^2^ = *σ_G_*^²^/(*σ_G_*^²^ + *σ_e_*²),(3)
where *σ_G_*² and *σ_e_*² are the genetic and residual variance, respectively.

We computed pairwise non-parametric Spearman’s rank correlation coefficients between the 68 scaled traits. We represented significant pairwise correlations using a 0.05 *p*-value threshold with the R packages *corrplot* and the ‘hclust’ clustering method.

Metabolomic profiles from each biological replicate were produced with the R packages *ComplexHeatmap* and *dendextend* on scaled datasets. Clusters for metabolites and accession profiles were produced with the R package *hclust* and the ‘average’ method on scaled dataset. Metabolic clusters were further validated with the *pvclust* R package based on ‘Euclidean’ distance, ‘average’ method, and n = 1000 bootstrap replications as explained in [25].

### 4.8. Trait and Gene Inheritance Mode

We compared the values of the 14 HF1 to their respective parent values for every trait to decipher trait inheritance pattern. Likewise, we compared candidate gene expression levels within each cross to assess gene inheritance. We first computed a one-way ANOVA to assess the genotype effect of gene and trait variation within each cross. For significant tests (*p* < 0.05), we estimated additive (A) and dominance (D) components of the genetic variation as described in [28]. Fruit weight, SSC, secondary metabolite and gene inheritance modes were thus classified as over-recessive (OR; D/A < −1.2), recessive (R; –1.2 ≤ D/A ≤ –0.8), additive (A; –0.8 < D/A < 0.8), dominant (D; 0.8 ≤ D/A ≤ 1.2), or over-dominant (OD; D/A > 1.2) in a given cross depending on the D/A ratio. We classified the trait or gene inheritance as ‘ns’ when no significant genotype effect was found within a cross. 

### 4.9. Identifying Maternal Effects for Traits and Genes

We computed Student’s *t*-Tests for each pair of reciprocal hybrids and each trait to identify maternal effects driving trait variation with a 0.05 *p*-value threshold. Maternal effects for gene expression were identified with the *DiCoExpress* pipeline, where each of the five pairs of reciprocal hybrid expression were compared.

## 5. Conclusions

The present analysis provides a detailed characterization of the modes of inheritance of tomato fruit volatiles and phenolics within 20 HF1 and their six parental lines covering a wide range of genetic variation. We report several conclusions that may help flavour and nutritional quality improvement in tomato modern varieties. First, we highlighted patterns in the mode of inheritance of metabolites: (i) transgressions were mostly of a single type for a metabolite whichever the cross; (ii) modes of inheritance were scarcely consistent for a metabolite between the 14 crosses, except for terpenoid-derived VOCs and benzenoid VOCs; (iii) metabolites belonging to the same cluster displayed similar modes of inheritance for a given cross. Knowing these patterns and focusing on key compounds where non-additive patterns may be exploited, breeders may screen their hybrids for the combination of alleles needed (i) in F1 hybrids outperforming their parents for the metabolites they are interested in; (ii) at markers associated to the metabolites in previous GWAS and linkage mapping analyses. 

The co-expression network we build using the DEGs, followed by correlation analysis with metabolites, provided information on candidate genes that might be targets for improving metabolite contents. Future perspectives lie in confirming these results by studying a larger experimental design where an expression GWAS combined with GWAS for metabolite content may further link the correlations herein highlighted between transcriptome and metabolome, in addition to identifying causal polymorphisms to be used in MAS: as a medium-term objective, the expensive quantification of VOCs that cannot be assessed routinely in a breeding program could be avoided with the development of efficient markers to screen in breeding material.

## Figures and Tables

**Figure 1 ijms-23-06163-f001:**
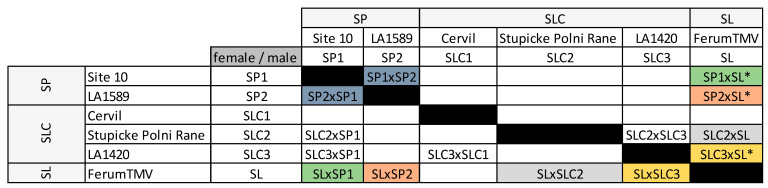
The six parental lines and 14 F1 hybrids (HF1) studied in the factorial design. Names are abbreviated to indicate the species the parental lines belong to. HF1 are named after the “female” × “male” they originate from. ‘*’ indicates that the HF1 was also studied in [25]. Pairs of reciprocal HF1 are represented in the same color. ‘SP’: *S. lycopersicum pimpinellifolium;* ‘SLC’: *S. lycopersicum* var. *cerasiforme*; ‘SL’: *S. lycopersicum* var. *lycopersicum*.

**Figure 2 ijms-23-06163-f002:**
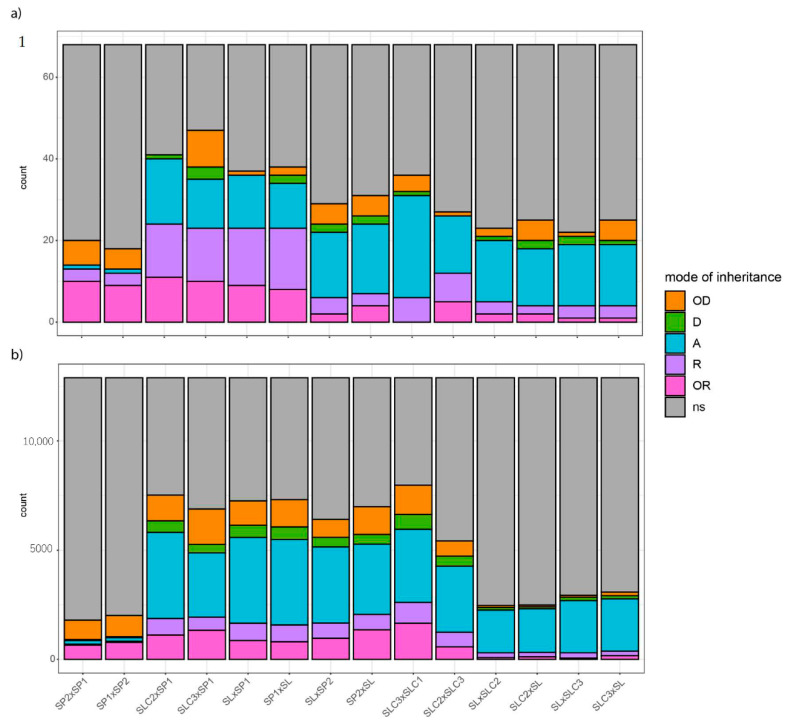
Overall modes of inheritance per cross for: (**a**) 66 metabolites, SSC and weight and (**b**) 12,896 differentially expressed genes (DEGs). The height and color of the barplot indicate the number of traits or DEGs with a specific mode of inheritance. Modes of inheritance are classified as over-recessive (‘OR’), recessive (‘R’), additive (‘A’), dominant (‘D’) or over-dominant (‘OD’) in a given cross. Traits or DEGs that show no significant difference (‘ns’) between at least two individuals of the cross (the two parental lines and HF1) are filled in grey.

**Figure 3 ijms-23-06163-f003:**
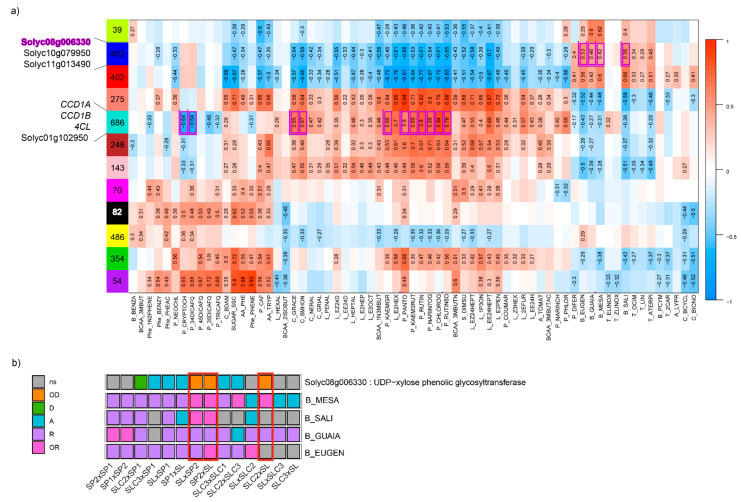
Integration of metabolome and transcriptome datasets by co-expression network analysis. (**a**) correlations between co-expression-modules and metabolites with the cloned (italic) and candidate genes found in co-expression modules significantly enriched in biological function linked to the synthesis of the metabolites correlated to the module. The number of differentially expressed genes (DEGs) within each module is indicated in the squares representing modules. Only significant correlations values are indicated. Positive correlations are indicated in red while negative correlations are indicated in blue; (**b**) focus on the mode of inheritance of the four benzenoid-derived VOCs (‘MESA’: methyl salicylate; ‘SALI’: salicylaldehyde; ‘GUAIA’: guaiacol; ‘EUGEN’: eugenol) in light of the mode of inheritance of the candidate gene Solyc08g006330 (coding for a glycosyltransferase). We circled in red the mode of inheritance of three crosses based on consistency between the gene and VOC modes of inheritance. Modes of inheritance are classified as over-recessive (‘OR’), recessive (‘R’), additive (‘A’), dominant (‘D’) or over-dominant (‘OD’) in a given cross. Traits or DEGs that show no significant difference (‘ns’) between at least two individuals of the cross (the two parental lines and F1 hybrids) are filled in grey.

**Figure 4 ijms-23-06163-f004:**
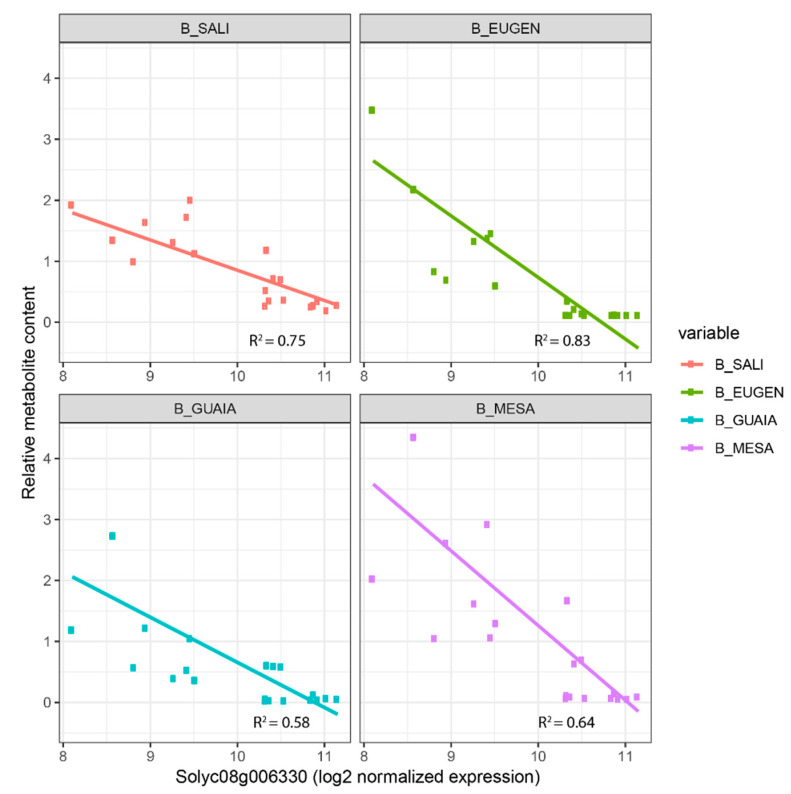
Benzenoid VOC contents according to the candidate gene Solyc08g006330 expression. We calculated coefficients of determination for each linear regression. ‘MESA’: methyl salicylate; ‘SALI’: salicylaldehyde; ‘GUAIA’: guaiacol; ‘EUGEN’: eugenol).

## Data Availability

The RNA-seq sequence data have been deposited in the sequence read archive (SRA) at NCBI under the accession number PRJNA532675 and are available at this address: https://www.ncbi.nlm.nih.gov/bioproject/PRJNA818106 (accessed on 27 April 2022). The following are available on the online dataverse https://data.inrae.fr/dataset.xhtml?persistentId=doi:10.57745/XOZA48 (accessed on 27 April 2022) including: the phenotype of the three biological replicates of the 20 genotypes studied within the factorial design for the 66 metabolites, SSC and fruit weight; the mode of inheritance of the 66 secondary metabolites, SSC and weight assessed within the 14 crosses; the normalized read counts of the 13,127 genes found expressed in the factorial design; the 12,896 DEGs identified in at least one contrast among the 13,127 genes found expressed in the factorial design; the mode of inheritance of the 12,896 DEGs.

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
