# Peer review of "Inheritance of Secondary Metabolites and Gene Expression Related to Tomato Fruit Quality"

_ijms, 2022, doi:10.3390/ijms23116163_

Round 1
Reviewer 1 Report
This manuscript is on the genetic expression for the quality in 14 F1 hybrid tomatoes.
It is stated that this manuscript was submitted in 2021. It may be necessary to check once again whether the submission has been duplicated in other journals.
The explanation of an abbreviation is usually not clear. When using an abbreviation for the first time, please write the full name before using the abbreviation.
Although it usually explains the results well, it is not clear which table or figure some results are presented in. Please make this a clearer.
The abbreviations used in the figures should be explained in detail (example: S. lycopersicum var. cerasiforme, 'SLC’) in the figure.
The current manuscript is so discursively written that it is possible to lose the gist of it while reading it through a series of too many characters. Thus, I think it will be easier for readers to understand this paper if it is provided in the figure of a schematic diagram that can contain the insightful contents of the authors' research conclusions.
Author Response
We would like to thank the reviewers for their positive remarks and tried to respond on a point-by-point bases to their specific remarks.
This manuscript is on the genetic expression for the quality in 14 F1 hybrid tomatoes.
It is stated that this manuscript was submitted in 2021. It may be necessary to check once again whether the submission has been duplicated in other journals.
- Sorry for the mistake, it was not submitted earlier, this is the first submission.
The explanation of an abbreviation is usually not clear. When using an abbreviation for the first time, please write the full name before using the abbreviation.
- We verified the manuscript, added full names before abbreviations and stated that the full names of secondary metabolites can be found in Table S1. Also, we defined abbreviations in each figure and table.
Although it usually explains the results well, it is not clear which table or figure some results are presented in. Please make this a clearer.
- We added more information as to the figures, supplementary figures, tables and supplementary tables to look at in the Results section.
The abbreviations used in the figures should be explained in detail (example: S. lycopersicum var. cerasiforme, 'SLC’) in the figure.
- We revised all figure captions to explain abbreviated terms and explain more thoroughfully the legend of all figures.
The current manuscript is so discursively written that it is possible to lose the gist of it while reading it through a series of too many characters. Thus, I think it will be easier for readers to understand this paper if it is provided in the figure of a schematic diagram that can contain the insightful contents of the authors' research conclusions.
- We wrote a conclusion in section 5 following the materials and methods (as asked in the template provided by IJMS) to provide better understanding of the main results and possible applications of this work.
Reviewer 2 Report
The manuscript "Inheritance of Key Compounds and Gene Expressions Related to Tomato Quality" by Estelle Bineau, José Luis Rambla, Renaud Duboscq, Marie-Noëlle Corre, Frédérique Bitton, Raphaël Lugan, Antonio Granell, Clémence Plissonneau, Mathilde Causse deals with the important interesting topic of gene regulation involved in various ways in the expression of the traditional taste of tomato. This work certainly deserves the interest of readers and tomato breeders and can be of great value for fresh fruits. The work can be accepted for publication, but it requires significant changes.
This manuscript contains no explicit purpose in the last paragraph of the Introduction. There is also no Conclusion section, which should clearly formulate conclusion and recommendations on how exactly breeders can use the proposed data.
The article is not framed according to the rules of MDPI. The drawings are of poor quality. Signatures are not visible, letters and digital images are small and illegible.
In addition, the figure captions do not contain complete information. The drawing should be a separate work in which everything is clear, all designations and abbreviations should be deciphered and unambiguous (see Figure 1-4).
In addition, the high-profile name is encouraging that it is possible to use the obtained valuable data in real breeding practice, but how to do this remains unclear. As a perspective, the option of continuing research with their expansion is considered, which indicates that the key indicated in the title of the article is not yet obvious.
I recommend changing the title of the article, correcting the goals and objectives from the description of the results to the answers to the question: why and how? Fix images and results to make them more explicit. Make changes to the discussion and formulate a conclusion, and then revise the abstract.
In this form, the article can not be considered in the IJMS journal.
Author Response
We would like to thank the reviewers for their positive remarks and tried to respond on a point-by-point bases to their specific remarks.
The manuscript "Inheritance of Key Compounds and Gene Expressions Related to Tomato Quality" by Estelle Bineau, José Luis Rambla, Renaud Duboscq, Marie-Noëlle Corre, Frédérique Bitton, Raphaël Lugan, Antonio Granell, Clémence Plissonneau, Mathilde Causse deals with the important interesting topic of gene regulation involved in various ways in the expression of the traditional taste of tomato. This work certainly deserves the interest of readers and tomato breeders and can be of great value for fresh fruits. The work can be accepted for publication, but it requires significant changes.
This manuscript contains no explicit purpose in the last paragraph of the Introduction.
 We corrected that
There is also no Conclusion section, which should clearly formulate conclusion and recommendations on how exactly breeders can use the proposed data.
- We wrote a conclusion in section 5 following the materials and methods (as asked in the template provided by IJMS) to provide better understanding of the main results and possible applications of this work.
The article is not framed according to the rules of MDPI. The drawings are of poor quality. Signatures are not visible, letters and digital images are small and illegible.
- Concerning the poor quality of the figures, we sent good quality pdf files of each figure individually but inserted .png figures in the text as it was required to submit the manuscript. Better quality can be found in a separate folder addressed to MDPI when the manuscript was submitted. We expect MDPI to integrate the figures in the .pdf quality if the manuscript were to be published after its revision.
In addition, the figure captions do not contain complete information. The drawing should be a separate work in which everything is clear, all designations and abbreviations should be deciphered and unambiguous (see Figure 1-4).
- We revised all captions (Figure 1-4) in order to specify abbreviations and to add additional layers of information. We hope that figures can now be understood without referring to the main text.
In addition, the high-profile name is encouraging that it is possible to use the obtained valuable data in real breeding practice, but how to do this remains unclear. As a perspective, the option of continuing research with their expansion is considered, which indicates that the key indicated in the title of the article is not yet obvious.
I recommend changing the title of the article,
- We removed “key” in the title and changed it to “Inheritance of Secondary Metabolites and Gene Expression Related to Tomato Quality”
Correcting the goals and objectives from the description of the results to the answers to the question: why and how? Fix images and results to make them more explicit. Make changes to the discussion and formulate a conclusion, and then revise the abstract.
In this form, the article can not be considered in the IJMS journal.
- We kept the discussion in its original form as these results need to be cross-checked with additional studies to be discussed more directly in the discussion part. However, we added a conclusion in section 5 following the materials and methods (as asked in the template provided by IJMS) to provide better understanding of the main results and possible applications of this work for breeders. We hope that this additional section may provide sufficient clarity compared to the previous version of the manuscript.
Reviewer 3 Report
Bineau et al. presented a very interesting (and appropriate for IJMS) research concerning the genetic inheritance of critical quality-related metabolites in tomatoes. The article is well written and the experiment is well performed, thus, I suggest that the article should be accepted after a minor revision.
The title is a bit confusing, consider revising it.
The organizing of the introduction should be vastly improved and shortened, to punctually provide the required background and highlight the scope of the research. As it is, the introduction is a bit unorganized and unfocused.
Line 37 “dedicated” is a bit awkward, I suggest to revise it
Line 40 revise 20th onward
Line 43 define HF1
Line 44 elaborate on “resistance genes”, revise “nowadays”
Line 45 define SLC, SL (or explain the required terminology before using it-please check similar use of unexplained terminology throughout the manuscript).
Lines 46-50 this phrase needs to be revise for improved consistency.
Line 48 revise “is on the right track”
Lines 67-68 revise for improved consistency, moreover, this phrase contradicts the statement above.
Line 109 revise
Lines 160-161 (and throughout the results part) the comparisons with published data usually belong to the discussion part.
Line 185 provide information about the consumer liking traits
Lines 186-187 revise
Author Response
We would like to thank the reviewers for their positive remarks and tried to respond on a point-by-point bases to their specific remarks.
Bineau et al. presented a very interesting (and appropriate for IJMS) research concerning the genetic inheritance of critical quality-related metabolites in tomatoes. The article is well written and the experiment is well performed, thus, I suggest that the article should be accepted after a minor revision.
The title is a bit confusing, consider revising it.
- We changed the title to “Inheritance of Secondary Metabolites and Gene Expression Related to Tomato Quality”
The organizing of the introduction should be vastly improved and shortened, to punctually provide the required background and highlight the scope of the research. As it is, the introduction is a bit unorganized and unfocused.
- We reorganized the introduction and shortened the background knowledge to give more information about the aim of this study.
Line 37 “dedicated” is a bit awkward, I suggest to revise it
- We changed to : More than 186 million tons were produced in 2020 [2], about 75% for the fresh market
Line 40 revise 20th onward
- We changed it starting from the 1950s
Line 43 define HF1
- It was done
Line 44 elaborate on “resistance genes”, revise “nowadays”
- We revised it to ‘pest and disease resistance genes’; revised in ‘F1 hybrids (‘HF1’), which represented 89% of commercialized varieties in Europe in 2018, combining up to eight resistance genes.’
Line 45 define SLC, SL (or explain the required terminology before using it-please check similar use of unexplained terminology throughout the manuscript).
- We checked for all abbreviations to be detailed when first used. SLC was already explained above (l.38-39)
Lines 46-50 this phrase needs to be revise for improved consistency.
- It was removed
Line 48 revise “is on the right track”
- It was removed
Lines 67-68 revise for improved consistency, moreover, this phrase contradicts the statement above.
- We changed the sentences so that the second statement directly follows the first for better understanding. “aroma, with no less than 400 volatile organic compounds (‘VOCs’) identified thus far in tomato [10]. The number of VOCs supposed to contribute to overall flavour has been cut down to approximately 30 [11]”
Line 109 revise
We changed the sentences to “They reported that most metabolites showed positive heterosis in the F1 hybrids, with up to 50% of mean increased content in the hybrids. On the other hand, agronomic traits were more subjected to additivity.”
Lines 160-161 (and throughout the results part) the comparisons with published data usually belong to the discussion part.
- We agree with this statement. However, we highlighted the consistency of our clusters with previous studies since we presented our results for modes of inheritance and metabolome-transcriptome integration in light of the metabolic pathway the metabolites belong to. So we chose to refer to bibliographic resources in the first part of the result to justify further comments on whole metabolic pathways (consistent with bibliographic resources) mode of inheritance / correlation to differentially expressed genes.
Line 185 provide information about the consumer liking traits
- We added in the first paragraph of the results section ‘The full names and abbreviations of the traits and secondary metabolites we assessed are available in Table S1. Table S1 also indicates which traits and metabolites impact tomato aroma.’
Lines 186-187 revise
- We changed it to “Phenylpropanoids (resp. flavones and flavonols) exhibited positive correlations between one another, but no significant correlation to flavones and flavonols (resp. phenylpropanoids) (Figure S1)”.